# Mapping electric fields and observation of ferroelectric domain switching in hafnia-zirconia devices by electron holography

Leifeng Zhang [1], Christophe Gatel [1,2] ✉, Muhammad Hamid Raza[3], Kilian Gruel[1], Catherine Dubourdieu [3,4] ✉ & Martin Hÿtch[1]

Ferroelectric devices such as capacitors, tunnel junctions and field-effect transistors rely on the reversible switching of polarisation under an electric field, which strongly depends on the screening charges at the interfaces. Despite the crucial role of charge trapping and detrapping on the performance of ferroelectric devices, current understanding relies heavily on electrical measurements of the whole device and/or local analysis of the atomic polarisation and structure. Here, we show how the internal electric fields can be measured within a ferroelectric $Hf_{0.5}Zr_{0.5}O_2/Al_2O_3$ tunnel junction using in situ electrical biasing electron holography. The charge densities at internal interfaces are quantitatively determined. Moreover, the polarisation switching of the ferroelectric film is mapped as the voltage gradually increases to the coercive voltage, revealing that switching occurs via both the nucleation and lateral growth of domains. This approach, complementary to existing techniques, opens new avenues for engineering the interfaces in ferroelectric devices.

Ferroelectric devices have a wide range of applications such as memories, transducers and transistors[1,2] and their physics is rich[3], including unique phenomena like negative capacitance[4,5]. The discovery of ferroelectricity in hafnia-based thin films[6] has revived the field of ferroelectric devices. In particular zirconium-doped hafnia (HZO)[7] has shown robust ferroelectricity at the nanometre scale[8–10], as well as excellent compatibility with complementary metal oxide semiconductor (CMOS) technology on Si substrate[10–12]. In this respect, there are three main types of ferroelectric devices which are being investigated with hafnia compounds: capacitors, tunnel junctions and field-effect transistors (FETs)[12,13]. They all rely on the ability to stabilise and reversibly switch by application of an electric field a ferroelectric polarisation.

All devices of interest to the semiconductor industry have thin dielectric layers at each interface of the ferroelectric layer, either because of the reactivity of the metal electrode (e.g., TiN or W) in air

and with the ferroelectric, or because the dielectric has been introduced on purpose, by design, such as in bilayer tunnel junctions[10,14] or FETs[15]. These ultrathin to thin dielectric layers play a crucial role in the electrical response of the device[16] and have to be carefully considered to optimise its performance. Indeed, their ability to provide screening charges at the interfaces determines the polarisation amplitude and its capacity to switch in both directions (perpendicularly to the interfaces). The amount of screening charges coming from the polarisability of the dielectrics is much smaller than the total amount needed to screen a ferroelectric polarisation of the order of few $\mu C\ cm^{-2}$ or tens of $\mu C\ cm^{-2}$. Hence, an additional source of screening charges has to be provided. When the dielectric is ultrathin (a few Å), the charges brought by the metal electrodes are the main source of screening charges. However, when the dielectric is a few nanometres thick, the screening has to be provided by the dielectric itself and this is where the charge trapping/detrapping plays a crucial role[17,18].

[1]CEMES-CNRS, Toulouse, France. [2]Université de Toulouse, Toulouse, France. [3]Helmholtz-Zentrum Berlin für Materialien und Energie, Berlin, Germany. [4]Physical and Theoretical Chemistry, Freie Universität Berlin, Berlin, Germany. ✉e-mail: christophe.gatel@cemes.fr; catherine.dubourdieu@helmholtz-berlin.de

Despite the crucial role of charge trapping and detrapping on the performance of ferroelectric devices[19], the only quantitative information on charges currently available comes from the modelling of electrical measurements[17,18,20,21] that have to make assumptions about their localisation. A direct technique that can unambiguously quantitatively map the charges and the local electric fields across the different layers of the stack is therefore lacking. Transmission electron microscopy (TEM) offers unprecedented spatial resolution for analysing the local polarisation and domain structure in ferroelectrics. Information on the electric dipoles at the atomic scale can be obtained by analysing the positions of atoms within the sample[22]. Such local observations can be correlated with the measurement of strain to study flexoelectricity[23] or multiferroic materials[24] and domain dynamics during in situ studies[25,26]. By visualising both heavy and light atoms, it was even possible to follow the structural phase change during switching and highlight the role of oxygen diffusion[27]. Nevertheless, such analyses remain structural and are not a direct measurement of the local electric dipole moment.

Four-dimensional scanning TEM (4D-STEM) aims to measure the electric field on the atomic scale[28–30]. It has been successfully used in ferroelectric thin films to study the local configuration of polarisation dipoles near interfaces[31,32] and in negative capacitance states[33]. Atomic electric fields are very strong, whereas the average electric fields in ferroelectric layers are much smaller. Here, 4D-STEM is not so accurate, and the results must take into account the variations in diffraction conditions associated with probe convergence[34].

To analyse in situ local electric fields and interface charging, we show that electron holography is most appropriate, particularly for the polycrystalline or amorphous materials studied here. We demonstrate that the average electric field in an individual ferroelectric layer can be mapped and measured. The technique is highly complementary to existing techniques as it provides the missing information required for a complete characterisation of ferroelectrics making the link between the macroscopic electrical properties and the local organisation of the atomic polarisation dipoles.

Electron holography is a powerful technique for quantitatively mapping local fields in materials, from electric and magnetic[35–40] to elastic strain[41]. Electric fields can be measured in semiconductors[42,43], microelectronic devices[44,45] and trapped charges can be mapped in high-$k$ materials whilst in situ biasing the sample[46,47]. It has also long been hoped that electron holography could measure the polarisation fields in ferroelectric materials, both at atomic and medium (nanometric) resolution[48]. The most convincing evidence came from the observation of stray fields around individual ferroelectric nanoparticles[49]. However, if the bounded charges generated by the polarisation in a metal-ferroelectric-metal capacitor are completely screened by free surface charges in the metallic electrodes, there is no depolarisation field and therefore electron holography experiments fail[50]. Another difficulty to detect ferroelectric polarisation is that the mean inner potential (MIP) of the material induces a much larger phase change than any ferroelectric effect, even during in situ biasing[51].

In this work, we measure the local electric field due to ferroelectric polarisation in a metal W/ferroelectric HZO/insulator $Al_2O_3$/metal W (MFIM) capacitor, where the depolarisation field is non-zero and we achieve the direct observation of polarisation switching in HZO on a large field of view of ∼300 nm. The quantitative mapping of internal electric fields is achieved by increasing the signal-to-noise ratio of the holography experiments through automation and, critically, by being able to switch the domains in situ. The following key points were controlled: (1) the ferroelectric layer must be set in a defined polarisation state in the starting experiment and (2) the sample has to be electrically cycled along its hysteresis curve to change the polarisation state in situ. The electrostatic phase component was measured for different polarisation states but otherwise identical conditions. In our approach, the background MIP and other static contributions, such as preparation artefacts, diffraction contrast and electron-beam induced charging can be subtracted. We thus show that the best way to achieve interpretable results is to compare directly holograms in two different polarisation states. The comparison allows us to follow the polarisation switching under an applied electric field. We demonstrate that the polarisation switching in HZO proceeds both by nucleation and lateral growth of domains. The capability to observe ferroelectric switching in situ across a wide field of view, to resolve the internal electric fields and to quantify trapped charge densities at individual interfaces provides new opportunities for engineering optimised ferroelectric interfaces to enhance nanodevice performance.

## Results
### Description of the ferroelectric tunnel junction

The ferroelectric tunnel junction (FTJ) was grown on a TiN bottom electrode with 9 nm of $Hf_{0.5}Zr_{0.5}O_2$ (HZO), a 3 nm barrier layer of $Al_2O_3$ and a top electrode of W (Fig. 1a). Similar MFIM stack devices were previously investigated as potential multilevel synaptic devices[10,18]. The current tunnels across the $Al_2O_3$ barrier when the HZO polarisation points towards the dielectric layer. When it points away from the $Al_2O_3$ barrier, the effective barrier width is large and block the tunnelling

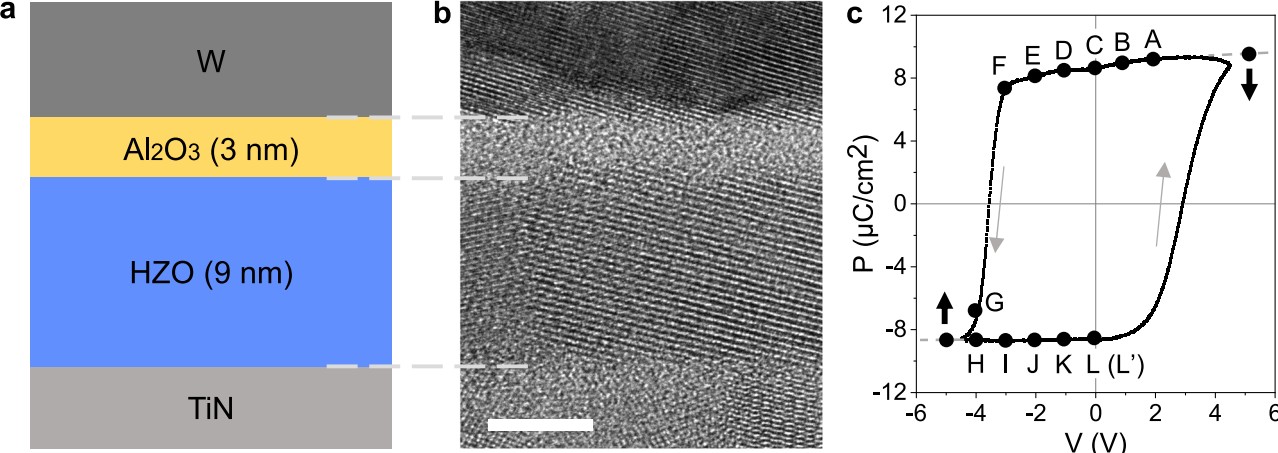

**Fig. 1 | Microstructure and hysteresis loop of designed FTJ device. a** Schematic map. **b** High-resolution TEM image. Scale bar is 5 nm. **c** Remanent P-V hysteresis curve measured by positive up negative down (PUND method) after 5000 wake-up cycles. In (**a**), HZO represents the $Hf_{0.5}Zr_{0.5}O_2$ layer. In (**c**), A, B, C, D, E, F, G, H, I, J, K and L (L'), respectively, represent the bias at +2 V, +1 V, 0 V, −1 V, −2 V, −3 V, −4 V, −4 V, −3 V, −2 V, −1 V and 0 V, in which L' points to 0 V from a 2nd cycle after −5 V.

current[10]. As shown by high-resolution TEM observation (Fig. 1b), the TiN, HZO and W layers are all polycrystalline whilst the $Al_2O_3$ layer is amorphous. Prior to the lamella preparation, the macroscopic capacitors ($95 \times 95\,\mu m^2$) were cycled (5000 cycles) in order to wake up the HZO layer. A typical hysteresis loop of the FTJ device is presented in Fig. 1c after wake-up (see also Supplementary Fig. S1). The remnant polarisation in HZO ($P_r = 8.5\,\mu C\,cm^{-2}$) is much lower than the one in metal/HZO/metal, typically of $\sim 17\,\mu C\,cm^{-2}$ for TiN/HZO/TiN and of $\sim 23$–$26\,\mu C\,cm^{-2}$ for W/HZO/W capacitors[52].

### Methodology of *operando* electron holography experiments

Specimens for *operando* electron holography experiments were prepared by focused ion beam (FIB), for a Hummingbird chip-based biasing holder (Supplementary Fig. S2). All observations were carried out on the I2TEM microscope (Hitachi HF3300-C) operating at 300 kV. The microscope setup also benefits from a direct electron detector, dynamic automation software for extremely long exposure times and smart acquisition routines which greatly improves the sensitivity of our measurements[53,54]. The spatial resolution was set at 0.7 nm on all phase images after data processing.

In the absence of magnetic fields, the phase shift of the electron hologram, $\phi$, is proportional to the electrostatic potential, $V$, encountered by the fast electrons along their trajectory:

$$\phi(x,y) = C_E \int V(x,y,z)dz \tag{1}$$

where $x,y$ are the directions in the image plane, $z$ the direction parallel to the electron beam and $C_E$ an interaction constant depending on universal constants and the accelerating voltage of the microscope ($C_E = 6.53 \times 10^6\,V^{-1}\,m^{-1}$ at 300 kV[55]). By far the largest contribution to the phase is from the MIP of the materials involved.

A bias of 5 V was first applied in situ to polarise the film in the down state (polarisation vector pointing towards the substrate) and electron holograms were then recorded at different applied biases following the remnant P-V curve shown in Fig. 1c. To determine the phase due to bias, we have adopted the following procedure: a hologram is acquired whilst the device is biased and another when both electrodes are grounded (0 V) keeping exactly the same conditions; the two hologram phases are then subtracted to remove all static contributions (see Methods). This procedure is key to subtract the MIP (Supplementary Figs. S12–S14,). Figure 2 shows phase images from recorded holograms at points A, B, D and E on the hysteresis curve (bias of 2 V, 1 V, −1 V and −2 V, respectively) after removing the phase from the hologram C (0 V). As expected, the phase due to the bias is uniform within the electrodes, with a positive phase difference across the insulating layers for positive bias and the opposite for negative bias (Fig. 2b, c). The phase shift between electrodes increases with the applied bias, and the sign changes as the bias is reversed.

The phase sensitivity can be estimated from the standard deviation of the variations in the Si substrate and found to be 45 mrad for 0.7 nm of spatial resolution. Close to defects notably, the crystal can diffract strongly and the fringe contrast is suppressed (see circled areas of the fringe amplitude in Fig. 2a). In these regions, all located in the top electrode of W, the phase is much less reliable. The region of interest in our study, i.e. the HZO and $Al_2O_3$ layers, presents a well-behaved phase variation without discontinuities.

To see the electric field induced by the polarisation, we subtract the phase of the grounded holograms in two different polarisation states, down (C) and up (L), but also in two similar polarisation states (L and L'). The L and L' holograms were recorded at 0 V after a 1st cycle and a complete 2nd cycle, respectively. As Fig. 3a clearly shows, there is a significant difference in the electrostatic phase within the insulating layers (HZO and $Al_2O_3$) between the up and down polarisation states. The phase difference between the two up states is close to zero

(Fig. 3b), thus ruling out any possible artefact. No change is also visible in the corresponding amplitude images, ruling out any contribution to the phase shift from variations in the diffraction conditions (Supplementary Fig. S13). The information is therefore purely electrical.

### Phase profile analysis in the FTJ stack

For a more quantitative comparison, line profiles across the device were extracted from the phase images (Fig. 3c, d) and converted into electrostatic potential using Eq. (1) considering the sample thickness ($69 \pm 3$ nm, see 'Methods'). Line profiles were averaged over 20 nm parallel to the interfaces to increase the phase precision from 45 mrad in the original phase image to 7 mrad in the phase profiles. We first performed extensive finite-element method (FEM) modelling, notably to quantify the small leakage fields above and below the sample (see Methods). The phase simulation reveals that only 5% of the signal comes from the stray field outside the sample. It is therefore possible to convert the measured phase shift directly into a potential, making only a small correction due to the leakage field in the error analysis. The simulated potential profiles are reported in red lines in Fig. 3c, d. The potential varies by only 0.2 V (0.1 rad) which would be masked in solely static experiments by the MIP contribution (>8 V) and points to the extreme sensitivity of our measurements.

A gradient in the electrostatic potential results in an electric field. In the electrodes, there is no electric field, as expected. However, when considering the difference between two opposite ferroelectric states (Fig. 3c), there is a clear gradient in the potential within the ferroelectric layer, indicating a positive (upward pointing) electric field. The strength of this electric field determined from the potential gradient was measured at $0.40 \pm 0.02\,MV\,cm^{-1}$ (see 'Methods') and is compensated by two opposing electric fields of $-0.64 \pm 0.12\,MV\,cm^{-1}$ in the $Al_2O_3$ layer and $-0.86 \pm 0.12\,MV\,cm^{-1}$ in an interfacial layer (IL) of about 2.2 nm between the TiN electrode and the HZO layer. The uncertainty in the local electric field is larger in the dielectric layers than in the HZO layer, essentially because both dielectric layers are thinner in width (see Methods) and it is more difficult to determine accurately their width if considering the projection effect along the electron path of an object around 70 nm thick. Despite this, the interfaces appear to have a low roughness.

This IL, clearly seen here on the electrical profile across the device, is not structurally visible in our correlative TEM and STEM characterisations (Supplementary Figs. S3–S5). However, electron energy loss spectrocopy (EELS) measurements performed using a probe-corrected microscope (see 'Methods') reveal the existence of an IL around 2 nm thick composed of Ti oxinitrides $TiO_xN_y$ ($\sim 1.5$ nm) with an increasing oxygen content and decreasing N content towards HZO, and eventually a $TiO_2$ layer ($\sim 0.5$ nm) with possibly some amount of Hf and Zr ($Ti_{1-x}Hf(Zr)_xO_2$) (see Supplementary Figs. S6–S8). Its presence is not surprising: TiN is known to oxidise very easily and to form oxinitrides and even $TiO_2$ when all N atoms are replaced by O atoms. The oxidation of the TiN bottom electrode is due to the air break between its sputtering deposition and the subsequent HZO deposition by atomic layer deposition (ALD), and also to the exposure to water during the first ALD cycles. A similar oxidation of TiN has been reported[56,57], for instance after an air break followed by ALD of HZO using ozone. Hard X-ray photoelectron spectroscopy (HAXPES) experiments, which allow to access buried interfaces, confirmed the EELS results (for these measurements the top W electrode is 5 nm-thick). The Ti *2p* and N *1s* core level spectra show the presence of Ti oxinitrides and a significant contribution from $TiO_2$ (or $Ti_{1-x}Hf_xO_2$)[58]. The Hf *4f* core level spectrum does not allow to evidence a possible Hf-O-Ti bond as the Hf *4f* doublet is quite insensitive to a change in second neighbours, as shown for thin films of the $HfO_2(Y_2O_3)$ solid solution[59]. $TiO_2$, as well as $Ti_{1-x}(Hf/Zr)_xO_2$, is a high-κ dielectric. Hence, it is likely the main contribution to the dielectric behaviour of the IL (together with the oxygen rich part of the oxynitride). Our highly sensitive holography experiments clearly demonstrate

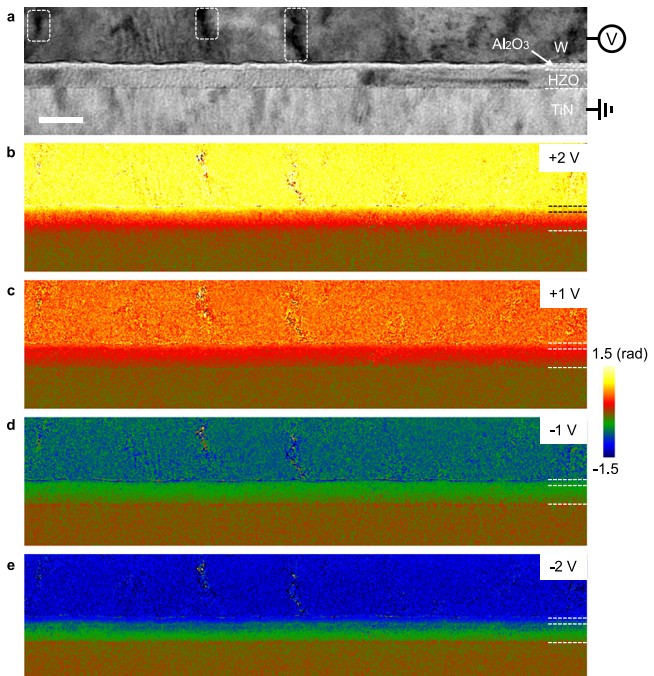

**Fig. 2 | *Operando* electron holography experiments on the studied bilayer FTJ device. a** Amplitude image showing the device architecture. The circled areas correspond to defects with strong diffraction effects where the fringe contrast is suppressed. Projected phase distribution maps showing electric potential at (**b**) +2 V, (**c**) +1 V, (**d**) −1 V and (**e**) −2 V after removing the phase contribution recorded at 0 V. Here, (**a**–**e**) correspond to the same region, and scale bar is 20 nm.

the electrical role of this interface layer and no longer leave any room for doubt about its influence on the device property.

### Quantification of charge trapped at the interfaces

In our case, it appears that charges are insufficient to completely screen the ferroelectric polarisation bound charges, which induces residual internal electric fields in HZO and dielectric layers. To quantify the number of charges at each interface, we also need to consider the interfacial layer between TiN and HZO in addition to the $Al_2O_3$ barrier.

Given the experimentally determined potential profile, the ferroelectric polarisation and interface charges can be determined. The number of free charges on the device per unit area, and hence the polarisation measured macroscopically, $P$, can be calculated through Gauss's Law:

$$P = \nabla \cdot \mathbf{D} = -D_{FE} \tag{2}$$

where $\mathbf{D}$ is the electrical displacement field, $D_{FE}$ is the electrical displacement associated with the ferroelectric layer. The sign assumes the top electrode is positively charged. This electrical displacement can be written:

$$D_{FE} = \varepsilon_0 \varepsilon_{FE} E_{FE} + P_{FE} \tag{3}$$

where $\varepsilon_{FE}$ is the background dielectric constant of the ferroelectric material, $E_{FE}$ the local electric field and $P_{FE}$ the spontaneous polarisation of the ferroelectric[60]. Combining these two equations gives the following result:

$$P_{FE} = -\varepsilon_0 \varepsilon_{FE} E_{FE} - P \tag{4}$$

We can therefore associate the experimentally measured local electric field ($0.40 \pm 0.02$ MV cm$^{-1}$) with the macroscopically measured

polarisation at remanence (8.5 μC cm$^{-2}$) to obtain an estimation of the spontaneous polarisation of the HZO layer to be $9.7 \pm 0.1$ μC cm$^{-2}$, assuming a relative permittivity of $32 \pm 2$ for HZO[18,61] (see 'Methods' for the error analysis).

We can also determine the charges trapped on the internal ferroelectric-dielectric layers following a similar reasoning and making use of Eq. (2):

$$\sigma^I = \nabla \cdot \mathbf{D} = D_{FE} - D_{DE} = -P - \varepsilon_0 \varepsilon_{DE} E_{DE} \tag{5}$$

Substituting for the local electric fields in the dielectric layers and assuming relative permittivities between 6 and 20 for the $TiO_xN_y$-$Ti_{1-x}(Hf/Zr)_xO_2$ interfacial layer[62] and $8 \pm 1$ for amorphous $Al_2O_3$[63], the trapped charges at the internal interface $TiO_xN_y$-$Ti_{1-x}(Hf/Zr)_xO_2$-HZO are estimated to be between −8 μC cm$^{-2}$ (permittivity of 6) and −7 μC cm$^{-2}$ (permittivity of 20). For the $Al_2O_3$-HZO interface, the result is more precise: $8.1 \pm 0.1$ μC cm$^{-2}$. The internal interfaces are indeed insufficiently charged to completely suppress the depolarisation field of the ferroelectric layer (Fig. 4).

### Observation of ferroelectric domain switching

As the residual electric fields related to the ferroelectric polarisation can be mapped and evaluated, we can analyse how the polarisation varies as a function of the applied external field when approaching the coercive voltage. Figure 5 shows the phase maps of the same region for various applied biases. As previously presented for the grounded states in Fig. 3, we subtracted the phases from a priori different polarisation states but for the same bias (D-K, E-J, F-I, G-H as marked bias points in Fig. 1c), removing the applied bias contribution and all other static contributions (MIP, beam charge effects, etc.).

Initially, at −1 V, the residual electric field appears to remain uniform in the layers (Fig. 5a) but from −2 V and particularly at −4 V, we see regions where the electric field disappears (Fig. 5b–d and Supplementary Fig. S9). The interpretation is that the polarisation state has flipped in these regions so that the subtracted holograms have the same phase. We therefore directly visualise the switching dynamics of the domains in the HZO layer through the residual electric field. It shows that the polarisation switches at different locations in the film in regions of about 10–15 nm in diameter (which is the typical size of the grains, see Supplementary Fig. S10) and that the domains of opposite polarisation grow laterally, propagating to the neighbouring grains as the voltage is increased while other localised nucleation keeps occurring as seen in Fig. 5c, d. Again, we observed no change in the diffraction contrasts in the amplitude images, regardless of the bias applied. The superposition of the grains in the lamella thickness does not allow to establish with certainty a relationship between the areas where the polarisation reverses first and particular crystallographic orientations. However, this is, to the best of our knowledge, the first visualisation of polar domain switching on such a large field of view in a HZO ferroelectric layer by in situ biasing TEM.

### Discussion

Polarisation switching mechanisms in $HfO_2$-based capacitors are extensively investigated and still debated. Switching in fluorite ferroelectrics such as polycrystalline HZO is commonly modelled using a nucleation-limited switching (NLS) model, which emphasises the role of localised nucleation events as the primary drivers of polarisation reversal[64]. This model suggests that switching occurs when local nucleation sites are activated, followed by limited domain growth (polycrystalline films offer many nucleation sites). The NLS model has been refined to take into account electric field inhomogeneities (recognising that different regions of the material experience varying field strengths due to microstructural variations) and to introduce time-varying nucleation rates to account for the dynamic nature of the switching under changing fields or as a result of nonuniformities in the

**a** (0V, ↓) - (0V, ↑)

0.5 (rad)

-0.5

**b** (0V, ↑) - (0V, ↑)

**c**

σ₁   σ₂

TiN | IL | HZO | Al₂O₃ | W

□□□□□ Exp.
▪▪▪▪▪ Sim.

**d**

TiN | IL | HZO | Al₂O₃ | W

□□□□□ Exp.
▪▪▪▪▪ Sim.

**Fig. 3 | Phase images, line profiles of projected electrostatic potential and FEM modelling simulations. a** Difference of phase images obtained in the same area under a bias of 0 V between the polarisation down (hologram C in Fig. 1c) and up (hologram L) of HZO. **b** Difference of phase images obtained on the same area than **a** under a bias of 0 V for two up states of the polarisation (holograms L and L') separated by a complete hysteresis cycle. **c** Electrostatic potential profile (in hollow grey squares) as well as simulated profile (in red solid dots) of the marked region in (**a**). **d** Electrostatic potential profile and simulated profile of the marked region in (**b**). IL is the interfacial layer between the TiN electrode and the HZO layer. To fit the phase profile in (**c**), two charging surfaces with charge densities noted σ₁ and σ₂ are placed at the IL-HZO interface and the HZO-Al₂O₃ interface, respectively. Line profiles were obtained by averaging the phase over 20 nm parallel to the interface. Scale bar is 20 nm.

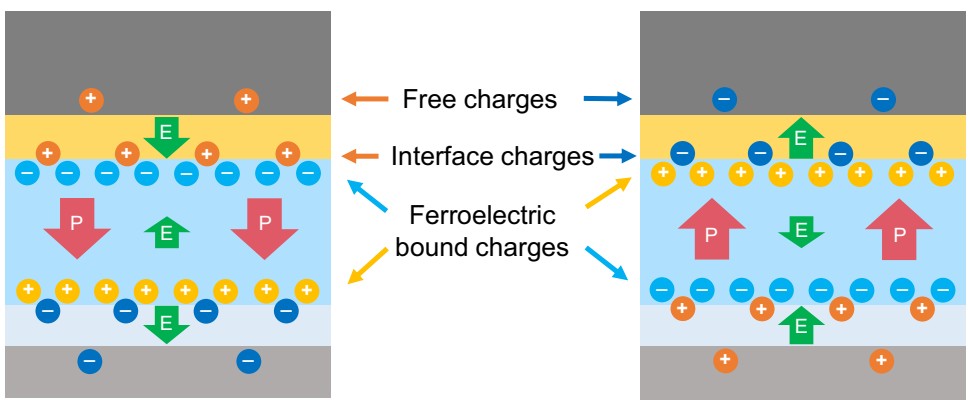

**Fig. 4 | Charge and electric field (E) distribution for both the down and up states of polarisation (P) in HZO.** Are represented free charges on electrodes, interface trapped charges, ferroelectric bound charges. Dielectric bound charges have been omitted. Charges on electrode and interface oppose ferroelectric bound charges but not completely, leaving a residual electric field.

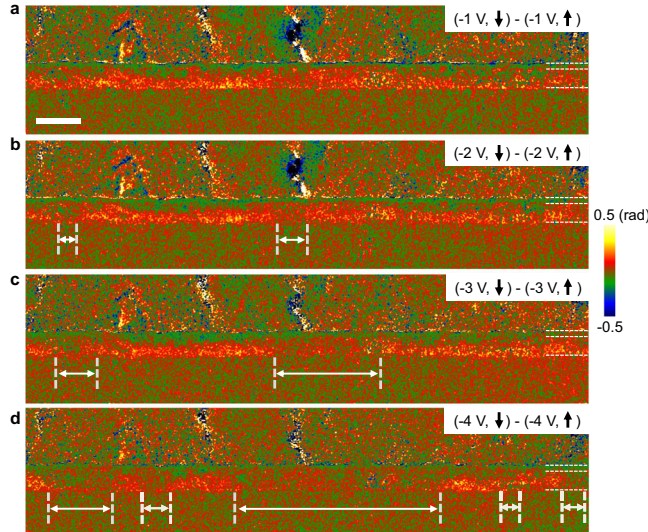

**Fig. 5 | Phase images highlighting the polarisation switching dynamics.**
**a** Difference of phase images under a bias of −1 V after switching the polarisation of HZO down (hologram D) then up (hologram K). **b** Difference of phase images (holograms E and J) at −2 V. **c** Difference of phase images (holograms F and I) at −3 V. **d** Difference of phase images (holograms G and H) at −4 V. All phase images were recorded in the same area and regions with white bidirectional arrows show the flipping of polarisation states. Scale bar is 20 nm.

film[65,66]. The NLS model and its variants have been successfully applied to HfO₂-based capacitors[67–74].

Domain wall (DW) motion/domain growth is usually thought to play a negligible role compared to nucleation, due to the polycrystalline structure, high defect density, small grain sizes (grain boundaries can act as pinning centres) and, most importantly, intrinsically large DW energy barrier as determined from first-principle calculations[75,76]. The common view of a negligible role of domain growth has been explained by the fact that orthorhombic HfO₂-based ferroelectrics have flat polar phonon bands leading to the localisation of the electric dipole in half of the unit cell while the other half is nonpolar[77]. In this picture, each localised dipole can, in principle, switch independently from the other ones, whether they are close or far from each other, thus limiting the role of domain growth. However, the involvement of domain propagation in HZO polarisation switching cannot be excluded. Choe et al. have proposed a new class of DWs (with a double-layer spacer) in the orthorhombic polar phase of HfO₂[78]. Their first-principle calculations show that these topological DWs exhibit energy barriers considerably lower than those calculated for the prevalent model[75,76], suggesting that the motion of these DWs after the nucleation of a switched domain is favourable, leading to the lateral growth of domains. Experimentally, using stroboscopic piezoresponse force microscopy, Buroghain et al. have shown that polarisation switching in polycrystalline Y:HfO₂ capacitors proceeds by both nucleation and lateral domain expansion[79].

The various experimental and theoretical results on the predominant mechanisms for polarisation reversal are still the subject of debate, and therefore, the direct observation of polar domains in HfO₂ compounds is extremely valuable. Note that crystalline domains and ferroelectric domains usually differ in size. Ferroelectric domains can be (and usually are) smaller in size than a crystalline grain (columnar grains are typically of 10−30 nm in diameter for our ALD-grown HZO films, see Supplementary Fig. S10). High-angle annular dark-field (HAADF) STEM studies, giving access to the atomically resolved positions of the projected Hf atoms, have been reported on HZO and other HfO₂-related nanocapacitors. These studies are conducted on a limited

field of view (typically 10 nm or below) inherent to the extreme high resolution needed for atomic position determination. Polarisation can be inferred but no information is gained on the ferroelectric domains at a large scale and on their dynamics.

As far as in situ STEM/TEM studies are concerned, they have focused on studying structural changes at atomic scale. Nukala et al. reported the structural study of epitaxial HZO/ La₀.₆₇Sr₀.₃₃MnO₃ (LSMO) heterostructures upon heating under vacuum by in situ STEM using integrated differential phase contrast (iDPC)[80]. Zheng et al. studied, at the atomic level, the first-order phase transition of polycrystalline orthorhombic HZO sandwiched between TiN electrodes, upon heating and cooling by in situ TEM[81]. In these studies, crystalline domains are determined but no information is gained on polarisation (and electric fields) and on polarisation reversal under an electric field.

In situ electrical biasing TEM studies of ferroelectric HfO₂ capacitors are even rarer. Nukala et al. reported a study based on HAADF and iDPC STEM imaging of epitaxial LSMO/HZO/LSMO capacitors grown on Nb-doped SrTiO₃ (the pristine epitaxial HZO is in the polar rhombohedral phase)[27]. They gained important information—at a local scale—on the crystalline structure with atomic scale precision. The visualisation of oxygen atoms allowed them to evidence a reversible migration of oxygen vacancies through the HZO layer and the associated phase transitions induced in the LSMO and HZO layers. However, no information is gained on the electrical potentials in the capacitor and their change upon application of an electric field. In our experiments, we do not access information on mechanisms at an atomic scale, such as oxygen vacancy movement, but study the polarisation reversal from a pure electrical point of view, and on large field of view of ~300 nm. This approach appears very complementary to HAADF and iDPC STEM.

Regarding the polarisation reversal mechanism, we observe that domains that have nucleated at −2 V with an opposite direction (Fig. 5b) grow laterally when the field is increased (Fig. 5c), with no nucleation events observed at −3 V (in the limit of detection) showing thus that DW propagation cannot be disregarded and play a significant role in the polarisation reversal. As the bias amplitude is increased to −4 V, other domains nucleate (Fig. 5d) and existing domains continue to propagate laterally. We would need of course, much more statistics to definitively conclude on the relative importance of nucleation and DW motion (which is beyond the scope of this study), but these results clearly show that domain growth, under DC field, plays a significant role in polarisation reversal.

In addition to accessing electrical fields, we gain information on the charges at the interfaces, in particular between the dielectric Al₂O₃ and the ferroelectric HZO. The ability to localise positive and negative charges and to quantify them is key for understanding the properties of industry-relevant FTJ and ferroelectric-FET (Fe-FET) devices. In Fe-FETs, charge trapping induces a shift of the threshold voltage $V_t$ opposite to the shift caused by the ferroelectric switching, therefore reducing the memory window. Trapped charges are also thought to be a cause of limited endurance, but, on the other hand, since they decrease the depolarisation field, they improve data retention[17,82–88]. In Fe-FET and FTJ devices, the ferroelectric HfO₂-based oxide is in contact with a thin dielectric (SiO₂ or SiON for Fe-FETs, and typically Al₂O₃ for FTJs), which leads to a lack of screening charges to compensate the bound charges associated to the polarisation. Simulations show, both for FeFETs and FTJs, that trapped charges help to stabilise and reverse the polarisation[18,20,89]. An optimal charge trap is required to maximise the tunnelling electro-resistance (TER) in FTJs, as shown in Fontanini et al.[18]. Whether the Fe-FETs and the FTJs are intended as memories or as memristive devices, the interplay between polarisation and charge traps is key for the device operation. Charge trap physical localisation and densities are determined by the material choice and by the processing conditions; accessing this information in a device is an important asset towards its optimisation.

In summary, we have reported the direct measurement of local electric fields within a ferroelectric stack and determined the density of interface charges. The influence of interfacial dielectric layers between the electrodes and the ferroelectric layer is unambiguously established. The oxidation of the bottom TiN electrode, leading to an oxynitride and a Ti(Hf/Zr)O$_2$ interfacial layer during processing is often disregarded in device modelling. We clearly show that such unwanted interfacial layer plays a significant role in the electric field profile and charge distribution and that it should be considered in device design and modelling. The charges that screen the depolarisation field are in majority located at the interfaces between the ferroelectric and the dielectric materials. We observed the polarisation switching in the HZO layer from a purely electrical point of view, by mapping at the nanoscale the residual electric field resulting from a non-total compensation of the interfacial charges. This switching proceeds both by a nucleation and growth mechanism.

We would like to emphasise the experimental challenges inherent to in situ holography of ferroelectric devices. Addressing these challenges required specific methodological developments to overcome technical limitations and establish a robust protocol, in which both sample preparation and the instrumental chain played a crucial role. In particular, the sample geometry had to provide a nearby field-free reference region while allowing the application of sufficiently high voltages to the thinned lamella for reliable polarisation switching. To ensure reproducibility, the HZO layer was cycled several times from a well-defined initial polarisation state. Although phase maps acquired at intermediate biases enabled direct visualisation of polarisation reversal, they required prolonged observation of the same area. Furthermore, additional field measurements at multiple locations within the layer were necessary to achieve a reliable statistical error analysis. The precision of 15 mV at a spatial resolution of 0.7 nm in the phase profiles in the growth direction results from averaging over 20 nm parallel to the layers. The error is higher than in the vacuum and is caused mainly by surface damage layers and contamination. The average electric field in the HZO layer could be measured to within 0.02 MV cm$^{-1}$ when averaged over 280 nm parallel to the layers. Here, the error comes mainly from the experimental uncertainties linked to the specimen thickness and exact magnification.

The developed methodology in this work demonstrates that in situ electron holography is a valuable technique for understanding the behaviour of ferroelectric devices, as it bridges the gap between electrical characterisation and atomic polarisation measurements, providing the vital information about local electric fields and charge screening phenomena with a sub-nanometric spatial resolution. The ability to determine the internal electric field in a large field of view and to quantify the trapped charge densities at each of the interfaces open avenues to engineering proper interfaces with the ferroelectric layer to optimise nanodevice performances.

## Methods
### Material elaboration and devices
A 30-nm-thick TiN layer was sputtered on p++ Si substrate, then a nominally 10-nm Hf$_{0.5}$Zr$_{0.5}$O$_2$ (HZO) film followed by a 3-nm Al$_2$O$_3$ layer, was deposited by ALD using 'Oxford FlexAl' system at 250 °C. Tetrakis(ethylmethylamino)hafnium (TEMA-Hf), TEMA-Zr and trimethylaluminum (TMA) were used as metal sources for HZO and Al$_2$O$_3$, respectively. Water was used as an oxidant for both ALD processes. W top electrodes were patterned (95 μm × 95 μm) by performing photolithography and lift-off of 30-nm W layer. A rapid thermal annealing was then realised at 400 °C for 120 s in N$_2$ atmosphere in order to crystallise the polar HZO phase. The low temperature crystallisation of HZO films renders the FTJ devices CMOS back-end-of-line compatible and enables three-dimensional (3D) integration with CMOS circuits[9,10].

### Electrical measurement
The electrical characterisation of the TiN/HZO/Al$_2$O$_3$/W FTJ devices was performed using a Keysight B1500A analyzer on an MPI TS2000-SE probe station. Prior to polarisation-voltage (P-V) and current-voltage (I-V) measurements, the fabricated FTJ devices were electrically cycled (wake-up) through 5000 cycles of a triangular waveform at 100 Hz with a ±4.5 V amplitude. The remanent polarisation was determined using the positive up negative down (PUND) method (triangular waveform, 1 kHz, ±4.5 V amplitude), which is accurate in the presence of leakage currents (see Supplementary Information). During *operando* electron holography experiments, the DC biasing was provided by a Keithley 2635B source metre and controlled by an in-house software.

### FIB-assisted nanodevice fabrication
Dedicated sample preparation is key to *operando* electrical biasing TEM experiments. Specimens for *operando* electron holography experiments were prepared by focused ion beam (FIB, Ga$^+$ source, Helios 600i from FEI), by which a lamella device is constructed and adapted on a commercialised Hummingbird chip compatible with biasing holder (1600 series, Hummingbird Scientific). The sample preparation was finished with a cleaning at low energy to minimise damage layers on the surfaces. For TEM/STEM characterisations, standard lamellas were prepared by typical FIB protocols.

### TEM/STEM characterisations
Bright-field (BF) STEM, high-angle annular dark-field (HAADF) STEM imaging, and STEM-EELS analyses were carried out on a JEOL JEM-ARM200F microscope operated at 200 kV and equipped with a CEOS ASCOR probe aberration corrector. EELS data were acquired using a Gatan GIF Quantum ER imaging filter coupled to a Gatan UltraScan 1000XP (model 994) camera. Two-dimensional EELS spectrum-imaging was performed with a 5-mm entrance aperture and a wide dispersion of 1 eV/channel, enabling the simultaneous detection of the W-M$_{4,5}$, Hf-M$_{4,5}$, Al-K, Ti-L$_{2,3}$, N-K, and O-K edges in the core-loss region. The energy resolution was approximately 3 eV. The spatial sampling step was 0.20–0.25 nm, corresponding to an uncertainty of 0.4–0.5 nm.

The high-resolution (HR) TEM imaging was conducted on a Hitachi HF3300-C (I2TEM) microscope operating at 300 kV. It is equipped with a cold-field emission gun (CFEG) for optimal brightness, a high-speed K3 camera (direct electron detector, model 1025, Gatan Inc.) and an imaging aberration corrector (BCOR, CEOS) allowing a spatial resolution of 0.08 nm.

### *Operando* electron holography experiments
*Operando* electron holography experiments were also performed on the I2TEM microscope, specially designed for electron holography and in situ experiments. The microscope is also equipped with a double stage configuration consisting of an upper stage positioned above the objective lens to allow the specimen to be observed in field-free conditions (Lorentz mode) and a conventional stage between the pole-pieces of the objective lens (normal mode). Owing to the aberration corrector, the Lorentz stage mode enables a spatial resolution of 0.5 nm and allows to reach a large field of view[90] that encompassed the substrate, bottom electrode, dielectric and ferroelectric layers, top electrodes and vacuum. Experiments were performed at an operating voltage of 300 kV, Lorentz stage, elliptical illumination and 2 post-specimen biprisms to allow flexibility in the holographic configurations and to eliminate the Fresnel fringe artefacts[91].

Holograms were recorded using a direct electron detection camera (K3 from Gatan Inc.) and with an interfringe of 0.45 nm (6.5 pixels). A total exposure time of 90 s associated the π - shift method[55] was set for each hologram thanks to dynamic automation[54]

(TEMControl) performed using different software developed: in-house FringeControl for feedback control of the holographic fringes, SpecimenControl for the specimen drift compensation (real time stage correction) and a specific process for the image acquisition.

Post-processing of the holograms, notably to extract the amplitude and phase images, was performed using software dedicated to quantitative hologram analysis, qHolo 1.0 (HREM Research Inc.), a plug-in for the image processing package in Digital Micrograph for GMS 3.3 (Gatan Inc.). A cosine Fourier-space filter giving a spatial resolution of 0.7 nm was selected for the amplitude and phase images. To remove the projector and camera distortion-induced phase modulations, a dedicated reference hologram recorded in a vacuum area during an exposure time of 480 s (8 min) was used. With this setup (I2TEM and K3 camera) combined to dynamic automation, the resulting phase noise after all data treatment processes is lower than 7 mrad in the TiN layer when the phase profile across the layers is averaged laterally over 20 nm.

All amplitude and phase images were aligned for the different biases applied using in-house scripts before being subtracting (see Supplementary Figs. S12–S14). Particular care was taken in aligning the images (see Supplementary Figs. S15–S17).

### FEM modelling simulation

Finite element modelling (FEM) was performed in a 2D geometry using COMSOL Multiphysics 5.6, a software package specifically designed to study physical systems, particularly those involving coupled multiphysics interactions. A stationary solver was used to compute the electrostatic distribution inside and outside the specimen.

The simulated domain was a $3.5 \times 3.5\,\mu m$ square, spanning both the propagation direction of the fast electron and the direction perpendicular to the interfaces (see Supplementary Fig. S18). The model incorporates the specimen geometry (lamella thickness, individual layer lengths, width of top and bottom electrodes), the relative permittivities and interfacial charging layers. The geometry was defined from the amplitude image reconstructed from electron holography. The initial thickness estimate of ~66 nm, obtained from the MIP in Si, was refined to 69 nm during COMSOL simulations to match the phase amplitudes measured at +1 V and −1 V (see Supplementary Fig. S19).

A non-uniform meshing strategy was employed to optimise computational efficiency while accurately describing both the active region and the surrounding vacuum. The triangular mesh gradually decreases in size from the external boundaries toward the active area, ranging from a maximum of $0.3\,\mu m$ down to 0.3 nm around the dielectric layer. These values correspond to the straight-line distances between neighbouring nodes. Because the nodes are not arranged on a cubic grid, the effective spatial resolution of the simulated projected potential is finer than these mesh values. The entire model was enclosed within an infinite-element domain to emulate open-boundary conditions. The bottom electrode and the outer boundaries were grounded, while a fixed potential was applied to the top electrode. The electrostatic potential was then computed and integrated along the propagation direction to simulate the phase shift of the fast electron, following Eq. (1) in the main text.

### Error analysis

To estimate the error in the measurement of the electric field, the phase gradient was determined by linear regression in 20 adjacent regions within the HZO layer, 14 nm (100 pixels) along the layers by 7 nm in the growth direction, corresponding to Fig. 3c, d. The phase gradient was then converted to the value of the electric field using Eq. (1) and corrected for the stray field by reducing the value by 5% (see Supplementary Fig. S20). The results are reported on Supplementary Fig. S11. The error bar for an individual measurement is given by the standard deviation of the results, or 0.04 MV cm$^{-1}$. The uncertainty in the average value is a factor of $\sqrt{20}$ less, or 0.01 MV cm$^{-1}$. By taking into account the uncertainty in the lamella thickness (69 ± 3 nm) and the magnification (2%), the overall uncertainty is 0.02 MV cm$^{-1}$ for the remanent field (Fig. 3c) giving: 0.40 ± 0.02 MV cm$^{-1}$. The uncertainty in the electric field within the dielectric layers is much higher, given that they are much narrower. We estimate the electric fields to be −0.86 ± 0.12 MV cm$^{-1}$ and −0.64 ± 0.12 MV cm$^{-1}$ in the interfacial and alumina layers respectively.

For the error in the charge measurements, the uncertainty in the permittivity needs to be taken into account. We consider that the relative permittivity of the HZO layer to be 32 ± 2 and the alumina layer to be 8 ± 1. The interfacial layer is more complicated as the exact composition is not known and neither is it uniform. We estimate the effective relative permittivity for the amorphous layer to be between 6 and 20. The uncertainty of the ferroelectric polarisation (Eq. 4) only depends on the electric field and permittivity in the HZO layer and the macroscopically measured polarisation. The resulting value is remarkably precise at −9.7 ± 0.1 μC cm$^{-2}$ (down state). The charge density on the internal interfaces is more uncertain, given the larger uncertainty on the local electric field and permittivity. Combining these errors into Eq. 5, we can estimate the trapped charge densities between the interfacial layer and HZO to be between −8 μC cm$^{-2}$ (permittivity of 6) and −7 μC cm$^{-2}$ (permittivity of 20). For the alumina-HZO interface, the result is 8.1 ± 0.1 μC cm$^{-2}$.

## Data availability

The data that support the findings of this study are presented in the main text and the Supplementary Information. The holograms generated in this study have been deposited in the Zenodo database accessible from https://doi.org/10.5281/zenodo.17604737[92].

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

## Acknowledgements

The authors acknowledge funding of the French National Research Agency and of the Deutsche Forschungsgemeinschaft in the framework of the bilateral ANR/DFG project FEAT (ANR-19-CE24-0027-01/DFG 431399790) and Sylvie Schamm-Chardon for its coordination. The authors from CEMES-CNRS also acknowledge the French National Research Agency for the POLARYS project (ANR–23-CE42-0011) (C.G.) and the 'Investissement d'Avenir' program reference Nos. ANR–10-EQPX-38-01 and 11-IDEX-0002, the 'Conseil Regional Midi-Pyrénées' and the European FEDER-FSE for financial support within the CPER program. C.D. acknowledges funding from the European Union's Horizon research and innovation programme under grant agreement 101135398 (FIXIT) and funding by the European Union of the project number 101098216 (ERC Advanced Grant, LUCIOLE). Views and opinions expressed are, however, those of the authors only and do not necessarily reflect those of the European Union or the European Research Council Executive Agency. Neither the European Union nor the granting authority can be held responsible for them. We would also like to thank Etienne Snoeck for his careful proofreading of the manuscript.

## Author contributions

C.G., M.H. and C.D. initiated and supervised the study. M.H.R grew the samples and performed the macroscopic P-V measurements under the supervision of C.D. C.G. and L.Z. designed and developed the experimental procedure, L.Z. prepared the samples for electron holography and assisted C.G. in performing the in situ

experiments. L.Z. prepared the samples for high-resolution TEM/STEM analyses and performed the experiments. C.G. and L.Z. processed the electron holography data. L.F. and K.G. performed the FEM modelling. C.G., L.Z., K.G., C.D. and M.H. interpreted holography results and FEM modelling and discussed the results. M.H. verified the electrostatic calculations. M.H. and L.Z. wrote the first draft of the paper further completed and edited by C.D. and C.G. All authors contributed to the discussion of the results and the correction of the manuscript.

## Funding

## Competing interests

The authors declare no competing interests.
