## [Transparent Peer Review file · Nature Communications]

Mapping electric fields and observation of ferroelectric domain switching in hafnia-zirconia devices by electron holography

Corresponding Author: Dr Christophe Gatel

Version 0:

Reviewer comments:

Reviewer #1

(Remarks to the Author)

This manuscript reports an impressive study of ferroelectric $\text{Hf}_{0.5}\text{Zr}_{0.5}\text{O}_2$ tunnel junctions using in-situ electron holography, enabling quantitative mapping of internal electric fields and direct observation of domain switching. The approach successfully isolates polarization-related contributions from mean inner potential effects and provides measurements of residual fields and interfacial charge densities. These results clarify the critical influence of interfacial oxynitride/ TiO_2 layers on charge screening and device performance, bridging the gap between atomic-scale structural imaging and macroscopic electrical characterization. The observation of nucleation and lateral domain growth is interesting, as it provides new insight into switching mechanisms in hafnia-based ferroelectrics. The methodology is carefully executed, combining in situ holography with FEM modeling and complementary STEM/EELS analysis, and is likely to have a good impact on both fundamental ferroelectric physics and applied device engineering. I recommend publication after minor revisions with the following comments:

- Please clarify how potential artefacts from FIB thinning or beam effects were mitigated or excluded for this research.
- How do the observed nucleation and lateral growth dynamics compare with existing switching models?
- Please briefly state the limits of spatial/temporal resolution and sources of uncertainty to aid reproducibility by other groups.

Reviewer #2

(Remarks to the Author)

The manuscript, as previously reviewed by and transferred from Nature Nanotechnology, presents an in situ electron holography study of ferroelectric HZO devices to directly visualize internal electric fields and polarization switching. Using a $\text{W}/\text{HZO}/\text{Al}_2\text{O}_3/\text{W}$ MFIM structure, the authors were able to quantify interfacial charge densities and map domain evolution during electrical cycling. The results show that polarization reversal occurs through nucleation and lateral domain growth. This approach provides a direct, quantitative means to probe local electric fields and charge trapping in ferroelectric devices, offering valuable insights for interface engineering and performance optimization.

In the current version, the authors have taken previous review comments into account and revised the manuscript to better clarify the scope and objectives of this study, while appropriately acknowledging the remaining challenges that may not be entirely addressed by (and fall beyond its scope of) this work. Overall, the revised manuscript is technically sound, scientifically valuable, and suitable for the readership of Nature Communications.

In general, the manuscript is well written. However, a minor issue remains: quite a few SI figures are not referenced in the main text, which makes some related content appear disconnected and thus may hinder the reader's ability to follow key details. It is suggested to explicitly cite all relevant SI figures in the manuscript for improved clarity and coherence.

Version 1:

Reviewer comments:

Reviewer #1

(Remarks to the Author)

The authors have satisfactorily addressed all of my concerns, and the manuscript is ready for publication.

Reviewer #2

(Remarks to the Author)

The revised manuscript has addressed all previous comments and is recommended to accept for publication.

We read carefully the reviewer comments and we believe that we have taken all of them into account and followed their recommendations. Additional modifications in the manuscript as requested by the referees appear in red.

Yours faithfully

Christophe Gatel on behalf all authors.

For the reviewers

Answer to Reviewer #1:

This manuscript reports an impressive study of ferroelectric Hf_{0.5}Zr_{0.5}O₂ tunnel junctions using in-situ electron holography, enabling quantitative mapping of internal electric fields and direct observation of domain switching. The approach successfully isolates polarization-related contributions from mean inner potential effects and provides measurements of residual fields and interfacial charge densities. These results clarify the critical influence of interfacial oxynitride/TiO₂ layers on charge screening and device performance, bridging the gap between atomic-scale structural imaging and macroscopic electrical characterization. The observation of nucleation and lateral domain growth is interesting, as it provides new insight into switching mechanisms in hafnia-based ferroelectrics. The methodology is carefully executed, combining in situ holography with FEM modeling and complementary STEM/EELS analysis, and is likely to have a good impact on both fundamental ferroelectric physics and applied device engineering. I recommend publication after minor revisions with the following comments:

- Please clarify how potential artefacts from FIB thinning or beam effects were mitigated or excluded for this research.
- How do the observed nucleation and lateral growth dynamics compare with existing switching models?
- Please briefly state the limits of spatial/temporal resolution and sources of uncertainty to aid reproducibility by other groups.

We sincerely thank the referee for his/her positive assessment and recommendation for publication.

Below, we address the three points raised:

- FIB thinning artefacts and beam effects

Artefacts related to FIB thinning were minimized at the final stage of specimen preparation by performing a low-energy (1 kV) and low-current (28 pA) ion-beam cleaning step, in order to mitigate possible damage effects (as specified in the Methods section, page 13). Other artefacts, which remain constant with the applied bias, are inherently removed when the phase image acquired at 0 V (grounded electrodes) is subtracted from that obtained under applied voltage. Moreover, FEM simulations indicate that damaged surface layers have no significant influence on the electrical properties. Similarly, beam-induced effects are cancelled by subtracting the 0 V reference image acquired under identical imaging conditions. This methodology is introduced at the end of the Introduction (page 3)

and detailed in the Results section (bottom of page 4). We also verified that the specimen exhibited identical characteristics for different 0 V phase images, confirming its stability over time.

- Discussion on switching mechanisms

A detailed discussion on the mechanisms governing polarisation reversal is already provided in pages 10–12 of the manuscript. In summary, switching in fluorite ferroelectrics such as polycrystalline $\text{Hf}_{0.5}\text{Zr}_{0.5}\text{O}_2$ (HZO) is commonly described by the nucleation-limited switching (NLS) model⁶⁴, which attributes reversal to the activation of local nucleation sites followed by limited domain growth. This model has been refined to account for electric-field inhomogeneities and time-dependent nucleation rates^{65,66}, and successfully applied to various HfO_2 -based capacitors^{67,74}.

Domain-wall (DW) motion has generally been considered negligible due to the polycrystalline structure, high defect density, small grain size, and intrinsically large DW energy barrier predicted from first principles^{75,76}. However, a nice work⁷⁸ proposed a new class of topological DWs with double-layer spacers and significantly lower energy barriers, suggesting that lateral domain propagation may occur more readily than previously assumed. Stroboscopic piezoresponse force microscopy⁷⁹ further confirmed that switching in polycrystalline Y:HfO_2 capacitors proceeds via both nucleation and lateral domain expansion.

While atomic-resolution HAADF and iDPC-STEM studies^{27,80,83} have provided valuable insights into structural evolution and local phase transitions, they probe only limited areas (<10 nm) and do not provide information on large-scale ferroelectric domains or the internal electric fields driving switching. In our *in situ* electron holography experiments, we directly visualized the evolution of internal electric fields over ~300 nm fields of view. Domains nucleated at –2 V were observed to expand laterally with increasing bias, with no additional nucleation events detected at –3 V. At –4 V, new domains appeared while existing ones continued to grow. These findings demonstrate that, contrary to the prevailing view, DW propagation plays a significant role in the polarisation reversal of HZO under DC electric field.

- Limits of spatial/temporal resolution and sources of uncertainty

One of the main challenges of this work lies in preparing a specimen that allows *in situ* polarisation reversal without degrading the multilayer stack, as highlighted at the end of the Introduction (page 3). Optimizing the signal-to-noise ratio (SNR) is also essential to detect the small phase variations measured. This is achieved by combining long exposure times with the use of a direct electron detection camera (see page 4 and Methods). The spatial resolution (0.7 nm here) depends on the interference fringe spacing (0.45 nm / 6.5 pixels), the noise level within the hologram, and the desired field of view. Higher spatial resolution could be achieved, but at the cost of a smaller field of view, which would prevent the observation of nucleation and domain growth during switching.

The electrical potential was measured to a precision of 0.1 V at a spatial resolution of 0.7 nm. When averaged over 20 nm parallel to the interface in the phase profiles, the precision is 15 mV. The error is higher than in the vacuum and is caused mainly by the surface layers of the sample and contamination. The electric field in the HZO layer was measured to within 0.02 MV.cm⁻¹ when averaged over 7 nm in the growth direction and 280 nm parallel to the layers. Over half the error comes from the systematic errors linked

to the uncertainty in the specimen thickness and the exact magnification. Without modeling of the stray fields, the error would be larger again. At higher spatial resolution, within the 2.2 nm wide interface layers, the error was 0.14 MV.cm⁻¹ and comes mainly from the precision of the phase measurement rather than systematic errors. It is difficult to generalise the performance given that the errors are a combination of the sample quality and the experimental and modeling uncertainties.

Finally, the temporal dynamics are inherently limited by the exposure time required for acquiring one hologram (90 s – 120 s) to collect a sufficient number of electrons and maintain an adequate SNR. Therefore, the polarisation reversal is studied in a quasi-static regime. Pump–probe approaches (stroboscopic mode) could in the future enable access to faster temporal dynamics, provided that domain nucleation and growth occur reproducibly during each cycle.

We added the following paragraph at the end of the discussion (page 13):

“The precision of 15 mV at a spatial resolution of 0.7 nm in the phase profiles in the growth direction results from averaging over 20 nm parallel to the layers. The error is higher than in the vacuum and is caused mainly by surface damage layers and contamination. The average electric field in the HZO layer could be measured to within 0.02 MV.cm⁻¹ when averaged over 280 nm along the layers. Here, the error comes mainly from the experimental uncertainties linked to the specimen thickness and exact magnification. Generalising the performance to other systems is difficult given that the sample quality, data averaging, experiment and modeling uncertainties all play a role.”

Answer to Reviewer #2:

The manuscript, as previously reviewed by and transferred from Nature Nanotechnology, presents an in situ electron holography study of ferroelectric HZO devices to directly visualize internal electric fields and polarization switching. Using a W/HZO/Al₂O₃/W MFIM structure, the authors were able to quantify interfacial charge densities and map domain evolution during electrical cycling. The results show that polarization reversal occurs through nucleation and lateral domain growth. This approach provides a direct, quantitative means to probe local electric fields and charge trapping in ferroelectric devices, offering valuable insights for interface engineering and performance optimization.

In the current version, the authors have taken previous review comments into account and revised the manuscript to better clarify the scope and objectives of this study, while appropriately acknowledging the remaining challenges that may not be entirely addressed by (and fall beyond its scope of) this work. Overall, the revised manuscript is technically sound, scientifically valuable, and suitable for the readership of Nature Communications.

In general, the manuscript is well written. However, a minor issue remains: quite a few SI figures are not referenced in the main text, which makes some related content appear disconnected and thus may hinder the reader’s ability to follow key details. It is suggested to explicitly cite all relevant SI figures in the manuscript for improved clarity and coherence.

We thank the referee for these positive remarks and his/her recommendation for publication.

We have taken the referee's suggestion into account and verified that all SI figures are explicitly cited in the manuscript and methods section.

For the reviewers

Answer to Reviewer #1:

Reviewer #1 (Remarks to the Author):

The authors have satisfactorily addressed all of my concerns, and the manuscript is ready for publication.

We sincerely thank the referee for his/her recommendation for publication.

Answer to Reviewer #2:

Reviewer #2 (Remarks to the Author):

The revised manuscript has addressed all previous comments and is recommended to accept for publication.

We sincerely thank the referee for his/her recommendation for publication.